# State Space Model-based Classification of Major Depressive Disorder Across Multiple Imaging Sites

Shusheng Li
International Research Institute for Artifcial Intelligence,
Harbin Institute of Technology (Shenzhen)
Shenzhen, China
lishusheng@hit.edu.cn

Yang Bo
Tencent
Shenzhen, China
yangbo@tencent.com

Ting Ma
1. Electronics & Information Engineering School, Harbin Institute of Technology (Shenzhen)
2. Peng Cheng Laboratory
3. Guangdong Provincial Key Laboratory of Aerospace Communication and Networking Technology, Harbin Institute of Technology (Shenzhen)
4. International Research Institute for Artifcial Intelligence, Harbin Institute of Technology (Shenzhen)
Shenzhen, China
tma@hit.edu.cn

Chenfei Ye*
International Research Institute for Artifcial Intelligence,
Harbin Institute of Technology (Shenzhen)
Shenzhen, China
chenfei.ye@foxmail.com

## ABSTRACT

Major Depressive Disorder (MDD) is a prevalent psychiatric condition characterized by persistent sadness and cognitive impairments, with high recurrence rates. This paper presents a novel state space model to classify MDD using BOLD time-series data, a brain function indicator represented as a vector of fMRI signals in the time domain. The analysis is based on data from 1642 subjects, each with 140 timepoints, encompassing multiple imaging sites. We propose an innovative model that leverages Mamba, a state-space model, to capture long-term dependencies in multivariate time series data while maintaining linear scalability. The model exploits the unique properties of time series data to produce salient contextual cues at multiple scales, utilizing an integrated Mamba architecture to unify the handling of channel-mixing and channel-independence situations. This approach enables effective selection of contents for prediction against global and local contexts at different scales. Performance evaluation demonstrates an average classification accuracy of 69.91% for the entire dataset. This paper underscores the potential of the proposed model in improving diagnostic accuracy.

## KEYWORDS

Classification, Major depressive disorder, Time series, State space model

## 1 INTRODUCTION

Major depressive disorder (MDD) stands as a leading source of disability across the globe. The current clinical approach to diagnosing MDD is predominantly based on evaluating symptoms and behaviors. However, the variability in the symptoms presented by individuals with MDD often results in misdiagnosis and delayed treatment. In light of this, the development of objective and quantifiable biomarkers for MDD could offer significant advantages, such as enhancing our understanding of the disorder's underlying mechanisms and facilitating the creation of biologically-informed diagnostic and treatment strategies.

Resting-state functional magnetic resonance imaging (rs-fMRI) emerges as a potential biomarker, offering a precise measure of the brain's functionality in the context of psychiatric conditions. Extensive research utilizing rsfMRI has indicated that individuals with MDD display irregular brain activity in various cortical and subcortical areas, including the prefrontal cortex, insula, amygdala, precuneus, and hippocampus. The integration of machine learning techniques has expedited the evolution of neuroimaging biomarker analysis, shifting from generalized population-based conclusions to personalized predictions that could enhance individualized clinical decision-making.

While Graph Convolutional Networks (GCNs) have shown promise in capturing complex brain connectivity patterns, they face significant limitations in this context. GCNs struggle with scalability issues when handling large, multisite datasets due to increased computational complexity and memory demands. They are also primarily designed to capture local neighborhood information, which may overlook long-range dependencies critical in fMRI data analysis. Moreover, GCNs can suffer from over-smoothing as the network depth increases, leading to loss of distinct features necessary for effective classification. Constructing an optimal graph structure that accurately reflects functional brain connectivity is challenging and sensitive to noise, which can adversely affect the modelâĂŹs performance.

The innovative model introduced in this paper centers around Mamba, a selective scan state-space model that serves as its core inference engine. Mamba is designed to address the complexities of long-term dependencies in time series data, which is particularly relevant for classifying MDD using BOLD timeseries data from fMRI scans. By leveraging Mamba's unique properties, the model

excels in producing salient contextual cues at multiple scales and effectively handling both channel-mixing and channel-independence situations. This dual approach ensures that the model can adapt to the diverse and complex nature of fMRI data, enhancing the selection of relevant features for prediction against both global and local contexts at different scales.

The Mamba model offers several advantages over Transformer-based models [13], particularly in the context of time series data analysis for classifying MDD using BOLD timeseries data from fMRI scans. While Transformers have become the go-to models for various sequence-based tasks due to their ability to capture long-range dependencies using self-attention mechanisms, they suffer from significant limitations, especially in terms of computational efficiency and scalability. Transformers typically exhibit quadratic time complexity relative to the sequence length, which leads to substantial computational and memory requirements, making them less practical for long-term time series data and large datasets like those involved in fMRI studies. In contrast, Mamba leverages the strengths of state-space models to capture long-term dependencies with linear scalability. This efficiency is achieved through the use of state-space representations that maintain a compact form of temporal dependencies, allowing the model to handle very long sequences without the exponential growth in computational resources that Transformers require. As a result, Mamba can process large-scale fMRI datasets more efficiently, providing faster and more scalable analysis.

Resting-state functional MRI scans of 1642 participants (848 MDD vs. 794 healthy controls (HC) ) across 16 sites of Rest-meta-MDD consortium were collected [11]. Performance evaluation demonstrated that the proposed model achieves superior classification accuracy, highlighting its potential in enhancing diagnostic accuracy for MDD. By addressing the challenges of capturing long-term dependencies and achieving scalability, this paper underscores the potential of state-space models in advancing the classification and understanding of major depressive disorder. The findings pave the way for future research and clinical applications, aiming to improve diagnostic accuracy and treatment outcomes for individuals.

Our contributions are summarized as follows: 1) The proposed model leverages a state-of-the-art state-space model architecture, designed to efficiently capture long-term dependencies in multivariate time series data. This addresses the inherent challenges of representing complex temporal dynamics in fMRI data, providing a robust framework for accurate MDD classification. The proposed method's ability to handle long-range temporal dependencies with linear scalability distinguishes it from traditional models like Transformers, which suffer from quadratic time complexity and substantial computational requirements. 2) Innovative State-space Model Approach. The proposed model leverages the capabilities of State-space model to capture long-term dependencies in multivariate time series data. This approach addresses the challenges of representing complex temporal dynamics in fMRI data, offering a robust framework for accurate MDD classification across multiple imaging sites. By effectively capturing long-range temporal dependencies, the model improves the ability to identify patterns associated with MDD. 3) Scalability and Computational Efficiency. The proposed model maintains linear scalability and a small memory footprint,

making it computationally efficient. This approach not only improves the model's adaptability to large datasets but also facilitates its practical application in clinical settings, where computational resources may be limited.

## 2 RELATED WORKS

Major Depressive Disorder is a widespread and debilitating psychiatric condition characterized by persistent sadness, cognitive impairments, and various other symptoms that significantly affect an individual's quality of life. Accurate diagnosis and understanding of MDD are crucial for developing effective treatment strategies. Traditional machine learning approaches for classifying MDD from fMRI data often involve manual feature extraction, where specific features such as regional connectivity strengths or network metrics are derived from the fMRI data. Support Vector Machines (SVM), Random Forests, and logistic regression models have been commonly used to classify MDD based on these features [21–24]. While these methods have shown promise, they are limited by their reliance on handcrafted features, which may not capture the full complexity of the underlying brain activity. Furthermore, these models often struggle with the high dimensionality of fMRI data, leading to issues such as overfitting and poor generalization to new data.

In recent years, deep learning models have gained popularity for their ability to automatically extract relevant features from raw data. Convolutional Neural Networks (CNNs) and Recurrent Neural Networks (RNNs) have been applied to fMRI data, demonstrating improved performance over traditional methods [25–28]. CNNs are particularly effective at capturing spatial patterns in the brain, while RNNs are adept at modeling temporal dependencies. However, these models can be computationally intensive and require large amounts of data to train effectively.

State-space models offer an alternative approach by maintaining a compact representation of the temporal dynamics in the data. SSMs have been used in various applications, including genomics and speech recognition, due to their ability to handle long-range dependencies with linear scalability. The innovative use of selective scan SSMs, such as Mamba, provides an efficient framework for processing large-scale fMRI data. These models can capture both short-term and long-term dependencies, produce salient contextual cues at multiple scales.

## 3 METHOD

To enhance the accuracy and reliability of major depressive disorder (MDD) classification across diverse imaging sites, we introduce an innovative state space model (SSM)-based framework tailored for time series data analysis. Our approach integrates the strengths of SSMs, enabling robust handling of complex multi-site imaging datasets.

## 3.1 Model architecture

The proposed model architecture is built upon a state-space model framework, which is designed to classify MDD using BOLD time-series data (shown in Figure 1). The Mamba model captures long-term dependencies in the multivariate time series data while maintaining computational efficiency and scalability. This section provides an explanation of each component of the model architecture, including the detailed mechanisms of the model.

**Input Representation.** The input to the model is a matrix $\mathbf{X} \in \mathbb{R}^{N \times T}$, where $N$ represents the number of brain regions (channels), and $T$ represents the number of timepoints. Each element $x_{i,t}$ in the matrix $\mathbf{X}$ corresponds to the BOLD signal from the $i$-th brain region at the $t$-th timepoint. This matrix serves as the primary data structure for processing and analysis within the model.

**State-Space Model Integration.** State-space models represent the state of a system as a set of variables evolving over time. These models are particularly effective for capturing temporal dependencies and dynamic behaviors in time series data. The Mamba model integrates SSMs to model the temporal dynamics of BOLD signals efficiently.

The continuous-time state-space model is defined as follows:

$$\frac{d\mathbf{h}(t)}{dt} = \mathbf{A}\mathbf{h}(t) + \mathbf{B}\mathbf{u}(t), \mathbf{v}(t) = \mathbf{C}\mathbf{h}(t), \qquad (1)$$

where $\mathbf{h}(t) \in \mathbb{R}^N$ is the state vector, $\mathbf{u}(t) \in \mathbb{R}^D$ is the input vector, $\mathbf{v}(t) \in \mathbb{R}^D$ is the output vector, and $\mathbf{A}$, $\mathbf{B}$, and $\mathbf{C}$ are coefficient matrices that define the system dynamics. The state vector $\mathbf{h}(t)$ captures the hidden states of the system, evolving over time based on the input vector $\mathbf{u}(t)$.

For discrete-time implementation, the continuous-time model is discretized as follows:

$$\mathbf{h}_k = \overline{\mathbf{A}}\mathbf{h}_{k-1} + \overline{\mathbf{B}}\mathbf{u}_k, \mathbf{v}_k = \mathbf{C}\mathbf{h}_k, \qquad (2)$$

where $k$ denotes the discrete time steps, and $\overline{\mathbf{A}} = \exp(\Delta t\mathbf{A})$, $\overline{\mathbf{B}} = (\Delta t\mathbf{A})^{-1}(\exp(\Delta t\mathbf{A}) - \mathbf{I})\mathbf{B}$, with $\Delta t$ being the time step interval.

**Multi-Scale Contextual Cues.** To capture both global and local temporal patterns, the proposed model employs a multi-scale approach for feature extraction. This approach involves generating features at different temporal resolutions, enabling the model to leverage contextual information at multiple scales (in Figure 2). At the high-resolution level, the model processes the BOLD timeseries data with minimal downsampling, retaining detailed temporal information. This level is crucial for capturing fast, transient changes in brain activity that may be indicative of MDD. $\mathbf{X}^{(1)} = \text{HighRes}(\mathbf{X})$, where HighRes($\cdot$) denotes the high-resolution processing function that maintains the original temporal granularity of the data. At the low-resolution level, the model downsamples the BOLD timeseries data, capturing broader, slower-changing patterns. This level helps in identifying long-term trends and dependencies that are essential for understanding the overall dynamics of brain function. $\mathbf{X}^{(2)} = \text{LowRes}(\mathbf{X})$, where LowRes($\cdot$) denotes the low-resolution processing function that reduces the temporal resolution of the data by a factor of $r$. The multi-scale feature extraction process ensures that the model can effectively capture a wide range of temporal dependencies, from short-term fluctuations to long-term trends.

**Channel Mixing and Independence Handling.** Our model incorporates mechanisms for handling both channel mixing and channel independence to manage the high-dimensional nature of fMRI data effectively. Channel mixing involves combining information across multiple brain regions to capture inter-region dependencies. This approach is particularly useful when there are strong correlations between different brain regions, which can provide valuable insights into the functional connectivity associated with MDD. $\mathbf{X}_{\text{mix}} = \text{Mix}(\mathbf{X})$, where Mix($\cdot$) denotes the channel mixing function that aggregates information across channels.

Channel independence, on the other hand, focuses on the unique temporal dynamics of each region. This approach is beneficial when the regions exhibit distinct patterns of activity that are important for the classification task. $\mathbf{X}_{\text{ind}} = \text{Indep}(\mathbf{X})$, where Indep($\cdot$) denotes the channel independence function that processes each channel separately. The model dynamically switches between channel mixing and channel independence based on the characteristics of the data, ensuring optimal feature extraction for the classification task.

**Outer and Inner Mambas.** The outer Mambas operate on the high-resolution data $\mathbf{X}^{(1)}$, capturing detailed temporal patterns. Each outer Mamba module processes the input data through a series of linear projections, causal convolutions, and state-space transformations.

$$\mathbf{h}_k^{(1)} = \overline{\mathbf{A}}\mathbf{h}_{k-1}^{(1)} + \overline{\mathbf{B}}\mathbf{u}_k^{(1)}, \mathbf{v}_k^{(1)} = \mathbf{C}\mathbf{h}_k^{(1)}, \qquad (3)$$

where $\mathbf{u}_k^{(1)}$ is the input to the outer Mamba at time step $k$, and $\mathbf{h}_k^{(1)}$ is the hidden state.

The inner Mambas operate on the low-resolution data $\mathbf{X}^{(2)}$, capturing long-term trends and dependencies. Similar to the outer Mambas, each inner Mamba module processes the input data through linear projections, causal convolutions, and state-space transformations.

$$\mathbf{h}_k^{(2)} = \overline{\mathbf{A}}\mathbf{h}_{k-1}^{(2)} + \overline{\mathbf{B}}\mathbf{u}_k^{(2)}, \mathbf{v}_k^{(2)} = \mathbf{C}\mathbf{h}_k^{(2)}, \qquad (4)$$

where $\mathbf{u}_k^{(2)}$ is the input to the inner Mamba at time step $k$, and $\mathbf{h}_k^{(2)}$ is the hidden state.

**Integration of Mamba Outputs.** The outputs of the outer and inner Mambas are integrated to form a comprehensive representation of the input data, leveraging both high-resolution and low-resolution contexts.

$$\mathbf{V} = \text{Concat}(\mathbf{v}_k^{(1)}, \mathbf{v}_k^{(2)}), \qquad (5)$$

where Concat($\cdot$) denotes the concatenation function that combines the outputs of the Mamba modules.

**Formulation of the Mamba Modules.** The Mamba modules are designed to capture long-term dependencies and selective attention mechanisms within the BOLD timeseries data. Each Mamba module consists of the following key components:

- Linear Projections. Linear transformations are applied to the input data to project it into a higher-dimensional space, facilitating the capture of complex temporal patterns.

$$\mathbf{u}_k = \mathbf{W}_1\mathbf{x}_k + \mathbf{b}_1, \qquad (6)$$

  where $\mathbf{W}_1$ and $\mathbf{b}_1$ are learnable parameters.

- Causal Convolutions. Convolutional layers with causal padding are employed to ensure that the temporal dependencies are captured without introducing future information.

$$\mathbf{c}_k = \text{Conv1D}(\mathbf{u}_k), \qquad (7)$$

  where Conv1D($\cdot$) denotes the causal convolution operation.

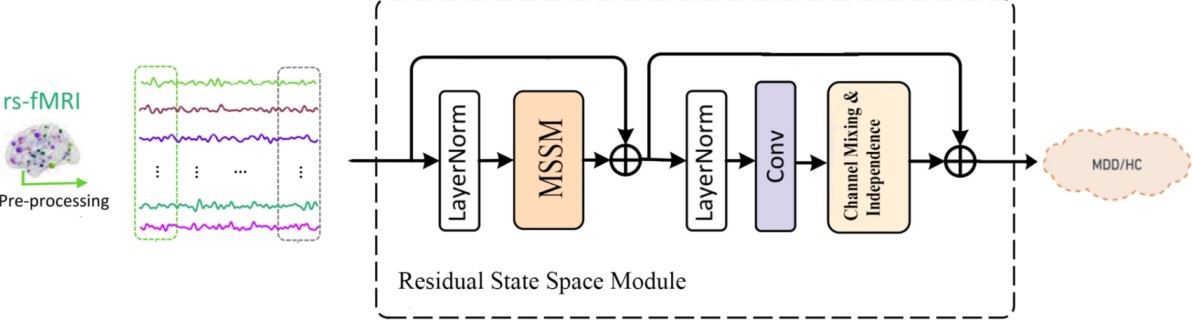

Figure 1: The overall network architecture of our method.Initially, rs-fMRI signals are preprocessed. Time series data are extracted and processed through multi-scale state space module (MSSM, as shown in Figure 2) to capture long-term dependencies. Outputs from the convolutional layers are through a residual connection and scaled. Finally, a fully connected linear layer transforms the features into classification logits, determining if the input data corresponds to MDD or HC.

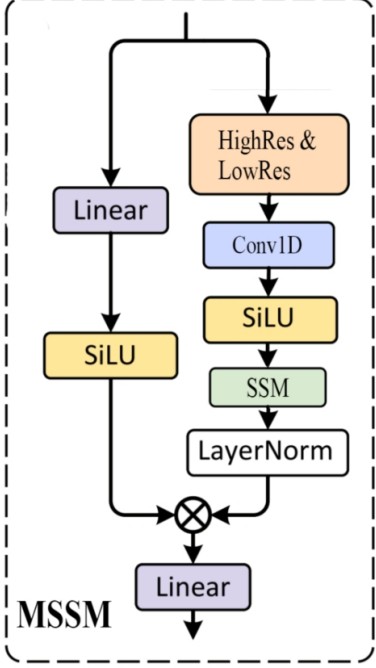

Figure 2: The overall network architecture of our method.Initially, rs-fMRI signals are preprocessed. Time series data are extracted and processed through multi-scale state space module (MSSM, as shown in Figure ??) to capture long-term dependencies. Outputs from the convolutional layers are through a residual connection and scaled. Finally, a fully connected linear layer transforms the features into classification logits, determining if the input data corresponds to MDD or HC.

- State-Space Transformations. The state-space model captures the temporal dynamics of the input data, with the hidden states evolving over time based on the input and previous hidden states.

$$\mathbf{h}_k = \overline{\mathbf{A}}\mathbf{h}_{k-1} + \overline{\mathbf{B}}\mathbf{c}_k, \mathbf{v}_k = \mathbf{C}\mathbf{h}_k, \tag{8}$$

- Selective Attention Mechanisms. The model incorporates attention mechanisms to selectively focus on relevant parts of the input data, enhancing the representation's discriminative power.

$$\mathbf{a}_k = \text{Attention}(\mathbf{v}_k), \tag{9}$$

where Attention($\cdot$) denotes the attention mechanism.

- Output Projections. The final output of each Mamba module is obtained through a linear projection, mapping the high-dimensional representation back to the original input space.

**Classification Layer.** The integrated features are passed through a classification layer to predict the presence of MDD.

$$\hat{\mathbf{y}} = \text{Softmax}(\mathbf{W}_3\mathbf{V} + \mathbf{b}_3), \tag{10}$$

where $\mathbf{W}_3$ and $\mathbf{b}_3$ are learnable parameters, and Softmax($\cdot$) is the softmax activation function.

## 3.2 Training and Optimization

The model is trained using a supervised learning approach with the cross-entropy loss function, which is suitable for classification tasks.

$$\mathcal{L} = -\frac{1}{N}\sum_{i=1}^{N}\left[y_i\log(\hat{y}_i) + (1-y_i)\log(1-\hat{y}_i)\right], \tag{11}$$

where $y_i$ is the true label, and $\hat{y}_i$ is the predicted probability of the $i$-th sample.

The Adam optimizer is used to update the model weights, providing efficient convergence.

$$\theta_{t+1} = \theta_t - \eta\frac{\hat{m}_t}{\sqrt{\hat{v}_t} + \epsilon}, \tag{12}$$

where $\theta$ represents the model parameters, $\eta$ is the learning rate, $\hat{m}_t$ and $\hat{v}_t$ are the bias-corrected first and second moment estimates, and $\epsilon$ is a small constant to prevent division by zero.

## 4 EXPERIMENTS

In this section, we conduct experiments to evaluate the performance of the proposed method.

### 4.1 Image acquisition and processing

Resting-state functional MRI and three-dimensional structural T1-weighted MRI images were collected from all participants at each local site. A standardized image preprocessing protocol was performed using the DPARSF toolbox. The preprocessing steps included slice timing correction, head motion correction, normalization, and removal of confounds as detailed in previous studies. For each subject, brain regions were partitioned using a brain atlas. The atlases used in this paper included the AAL-116 atlas [36] and the CC200 atlas [35], which yielded time series signals for 116 and 200 brain regions, respectively. Time series of BOLD signals from voxels in each ROI were extracted and averaged. Functional connectivity between each pair of ROIs was evaluated using the Pearson correlation coefficient of the corresponding time series. Fisher's z-transformation was then applied to the correlation estimates, yielding a $160 \times 160$ functional connectivity matrix for each participant.

### 4.2 Experimental Settings

Our study was performed based on 25 datasets from 17 hospitals in the Rest-meta-MDD consortium that included 848 MDD patients and 794 healthy controls. Demographic and clinical information including age, sex, illness duration, medication status, episode status, and 17-item Hamilton Depression Rating Scale (HAMD) were collected at each site.

In our method, we use the SiLU [29, 30] as activation functions and normalization layers (LayerNorm) [31]. The Adam optimization is applied with a learning rate of 0.001. The batch size is set to 32 and the number of epochs is set to 300.

### 4.3 Evaluation and performance metrics

We evaluate the performance of the proposed method with different methods. 80% of the data is used for model training and the remaining 20% as a validation set for testing. The parameters of the model are fine-tuned based on the results of the validation set, allowing us to obtain the best hyper-parameters. To comprehensively assess the performance of our method, three common metrics are used, including accuracy, precision, and recall.

### 4.4 Methods for Comparison

We compare our method with different. For fair comparison, these competing methods employ very similar learning schemes, as detailed below.

- 1DCNN: As a traditional deep learning method, the 1D Convolutional Neural Network (1DCNN) has been successfully applied in the field of computer vision. In this experiment, the output channels are set to 128, the kernel size is 2, and the stride is fixed at 1. To avoid the overfitting problem, we use batch normalization and maximum pooling.

**Table 1: Results of different methods in MDD VS. NC CLASSIFICATION**

| Method | Accuracy (%) | precision (%) | Recall (%) |
|---|---|---|---|
| 1DCNN | 58.72 | 59.62 | 67.52 |
| LSTM | 52.12 | 54.23 | 66.75 |
| 1DCNN_LSTM | 56.42 | 56.90 | 72.58 |
| ST-GCN | 51.12 | 47.54 | 60.24 |
| DKAN | 52.03 | 54.24 | 62.15 |
| Transformer-Encoder model | 67.21 | 68.60 | 63.96 |
| Ours | 69.91 | 70.62 | 67.96 |

- LSTM: The Long Short-Term Memory (LSTM) network is one of the most widely used classification models in neuroimaging analysis, demonstrating excellent performance in sequence data processing. In this experiment, the hidden layer size is set to 100, and there is one hidden layer. A fully connected layer is added after the LSTM to classify MDD.
- 1DCNN_LSTM: CNN and LSTM are widely used neural network layers, and their combination has been applied in numerous applications. In this experiment, a two-layer CNN is used, with the first layer having 256 channels and the second layer having 64 channels. The kernel size is set to 3, with a fixed stride of 1. The LSTM has a hidden layer size of 100 and consists of one layer. Following this, a fully connected layer is added to output the final classification results.
- ST-GCN: Graph convolution has demonstrated significant potential in capturing graph structures and has been applied to various functional connectivity (FC) analyses in psychiatric disorder studies. In this experiment, we utilized the ST-GCN model proposed by Azevedo *et al.* [32]. This model integrates Graph Convolutional Networks (GCN) with Temporal Convolutional Networks (TCN) to effectively learn features for classification from both the spatial and temporal components of resting-state fMRI (rs-fMRI) data in an end-to-end manner.
- DKAN: Zhang *et al.* [33] propose a diffusion kernel attention network (DKAN) that replaces the original dot product attention module in Transformers with kernel attention, significantly reducing the number of parameters. Additionally, this model employs a diffusion mechanism instead of the traditional attention mechanism, thereby enhancing the classification performance for mental disorders.
- Transformer-Encoder model: Dai *et al.* [34] propose a model based on Transformer-Encoder for MDD classification. The model discarded the Transformer's Decoder part, reducing the model's complexity and it does not require a complex feature selection process and achieves end-to-end classification.

Table 1 shows the performance of various methods for classifying MDD versus NC, highlighting differences in accuracy, precision, and recall. The proposed method achieves the highest accuracy at 69.91%, with a precision of 70.62% and a recall of 67.96%, indicating a well-balanced performance. LSTM, known for handling sequential data, underperforms with 52.12% accuracy, 54.23% precision,

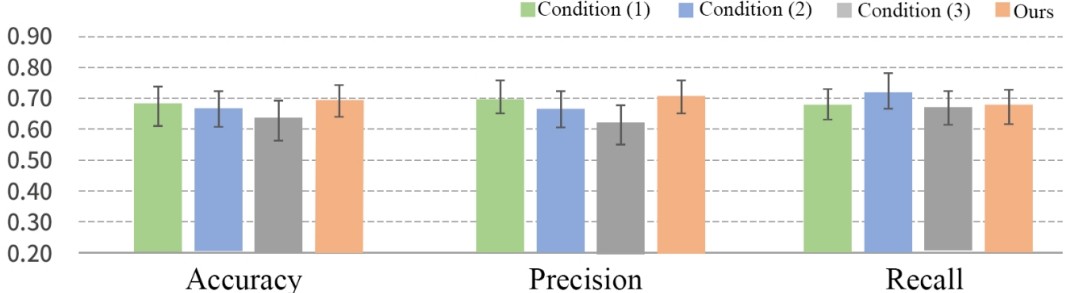

**Figure 3: Results of ablation study in MDD vs. NC classification.**

and 66.75% recall, suggesting it may not fully exploit spatial relationships in the data. The hybrid 1DCNN_LSTM model improves performance to 56.42% accuracy, 56.90% precision, and a high recall of 72.58%, effectively capturing both spatial and temporal features. The ST-GCN model, despite its advanced capability in handling spatial-temporal graphs, achieves lower results with 51.12% accuracy, possibly due to the complexity of graph structure learning or dataset-specific characteristics. The DKAN model, which introduces a kernel attention mechanism and diffusion process, achieves 52.03% accuracy, 54.24% precision, and 62.15% recall, indicating that while innovative, it may not be as effective in this classification task. The Transformer-Encoder model performs significantly better with 67.21% accuracy, 68.60% precision, and 63.96% recall, showcasing its strength in capturing long-range dependencies. Overall, the proposed method stands out with its superior balance of high accuracy, precision, and recall, demonstrating robustness and reliability in classifying MDD.

## 4.5 Ablation Study

In this section, we perform ablation experiments on the dataset and verify the effectiveness of the proposed model and method by AAL-116 brain atlas (in Figure 3) The ablation conditions included (1) only consider single-scale information (2) with and without residual connections and (3) whether to perform data augmentation.

The experimental results on ablation condition (1) indicate that the multi-scale setting achieved the best classification performance. This success can be attributed to the model's ability to capture features at different temporal resolutions, which is crucial for identifying patterns in complex time-series data. By using multiple convolutional layers with varying kernel sizes, the model can effectively learn both fine-grained and coarse-grained temporal features. This comprehensive feature extraction likely enhances the model's ability to distinguish between MDD and HC more accurately. This comparison underscores the importance that fuses multi-scale feature extraction with other model enhancements for optimal performance.

The experimental results of the ablation condition (2) show the effectiveness of residual connection. These residual connections help stabilize training and reduce overfitting, especially for the smaller datasets with channel independence.

The experimental results of ablation condition (3) indicate that the data augmentation strategy is not necessary in this study. The time series of each subject were randomly segmented into segments with a length of 90 time points using sliding windows. This simple data augmentation did not increase the diversity of the data, suggesting that more sophisticated augmentation techniques might be required to enhance model performance. The lack of improvement implies that the current dataset already possesses sufficient variability, and the applied augmentation did not add significant new information for the model to learn from.

## 5 CONCLUSION

This paper presents a novel state-space model for classifying MDD using BOLD time-series data from functional magnetic resonance imaging. The study involves a large dataset of 1642 subjects, collected from multiple imaging sites, ensuring diverse and comprehensive coverage. The proposed model leverages the strengths of SSMs to capture long-term dependencies in multivariate time series data while maintaining linear scalability and computational efficiency. it is designed to process multi-scale contextual cues, allowing it to handle both channel-mixing and channel-independence scenarios effectively. This dual approach ensures the model can adapt to the complex nature of the data, enhancing feature selection for prediction against global and local contexts at different scales. Performance evaluation shows that the proposed model achieves an average classification accuracy of 69.91% across the entire dataset, significantly improving diagnostic accuracy. The key contributions of this study include utilizing a large-scale MDD dataset, developing an innovative SSM-based classification approach, and ensuring scalability and computational efficiency. The findings underscore the potential of the proposed model in improving the classification and understanding of MDD, enhancing diagnostic accuracy and treatment outcomes for individuals with MDD.

## 6 ACKNOWLEDGMENTS

This study is supported by grants from the Guangdong Basic and Applied Basic Research Foundation (2023A1515010792, 2023B1515120065), National Natural Science Foundation of P.R. China (62106113), Basic Research Foundation of Shenzhen Science and Technology Stable Support Program (GXWD20231129121139001).

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
