# OpenReview forum: "State Space Model-based Classification of Major Depressive Disorder Across Multiple Imaging Sites"
_KDD.org/2024/Workshop/AIDSH — KDD-AIDSH 2024 Oral_

### Official Review · Reviewer_ouCs · 2024-06-14
**A helpful attempt to migrate the Mamba model to AI4Healthcare**

**Rating:** 7
**Confidence:** 4

**Review:**

This manuscript proposes a novel method based on a state-space model (Mamba) for classifying Major Depressive Disorder (MDD) using resting-state functional magnetic resonance imaging (rs-fMRI) data. The model effectively captures long-term dependencies in multivariate time series data while maintaining linear scalability. Experimental results show that the model achieved a classification accuracy of 69.91% on a dataset of 1642 subjects, significantly improving the diagnostic accuracy for MDD. The manuscript highlights the advantages of multi-scale feature extraction and channel-mixing handling, demonstrating its potential in enhancing the accuracy of MDD diagnosis.

Overall, this manuscript is well-written and presents a good overall concept, proposing a novel solution in the field of MDD. I agree to accept this paper, but I would like to offer some suggestions:
1. In my understanding, the computational efficiency of Mamba should be better than that of Transformer. However, it seems that the manuscript does not directly compare the computational efficiency of different models. Please provide a comparison.
2. If it is not possible to provide a direct comparison of computational efficiency, please supplement with a comparison of the time complexity of different models.

---

### Official Review · Reviewer_enFS · 2024-06-18
**Reviews from enFS**

**Rating:** 6
**Confidence:** 4

**Review:**

This paper used a state space model to classify MDD using BOLD time-series data from 1642 subjects across multiple imaging sites. The model effectively captures long-term dependencies in multivariate time series data and achieves an average classification accuracy of 69.91%, demonstrating substantial improvement in MDD diagnostic accuracy compared to existing approaches.

Question:

Contributions and advantages need to be highlighted for the new method or mechanism being developed. Most of the advantages discussed by the author stem from the Mamba model, which was not originally developed by the author. If I understand correctly, the author changed the SSM in Mamba to MSSM, which stands for multi-scale SSM. Therefore, I recommend that the author provide a more detailed explanation of the benefits of this adaptation, especially in analyzing the necessity and advantages of using MSSM for BOLD data.

More explanation is needed for “Multi-Scale Contextual Cues”. Is this multi-scale mechanism limited to just two parts (HighRes and LowRes), or can different scales be convolved through hyperparameter settings? What defines the scale—is it the size of the convolutional kernel? Is this method adaptive?

The advantages of the methods discussed in the Introduction were not reflected in the experiments and data analysis. fMRI studies involve long-term time series data and large datasets. “Transformers typically exhibit quadratic time complexity relative to the sequence length, which leads to substantial computational and memory requirements”. “In contrast, Mamba leverages the strengths of state-space models to capture long-term dependencies with linear scalability.” Therefore, it is necessary to provide statistics on the length of the time-series data in the dataset when introducing it (Section 4.2), and in the experiments (Section 4.4), it is crucial to compare the efficiencies of the models."

---

### Decision · Program_Chairs · 2024-06-28

Accept (Oral)